# Synthesis of Novel Li$_2$O-CuO-Bi$_2$O$_3$-B$_2$O$_3$ Glasses for Radiation Protection: An Experimental and Theoretical Study

**Maged S. Al-Fakeh** [1,2], **Emran Eisa Saleh** [3,4,*] and **Faisal Alresheedi** [5]

1   Department of Chemistry, College of Science, Qassim University, Buraidah 51452, Saudi Arabia
2   Taiz University, Taiz 3086, Yemen
3   Physics Department, Faculty of Science, University of Aden, Aden P.O. Box 6312, Yemen
4   Physics Department, Faculty of Science, Ain Shams University, Cairo 11566, Egypt
5   Department of Physics, College of Science, Qassim University, Buraidah 51452, Saudi Arabia
*   Correspondence: eesas2009@yahoo.com; Tel.: +966-55-7864972

**Abstract:** Glass samples were synthesized according to 10Li$_2$O + 20CuO + xBi$_2$O$_3$ + (70 − x)B$_2$O$_3$, where x = 0, 10, 20, 30, 40 mol% by the melt-quenching method. The ability of the prepared glass to protect against gamma rays and neutrons was examined experimentally and theoretically. The mass attenuation coefficient (MAC) was calculated experimentally at energies of 0.662, 1.173, and 1.333 MeV using $^{137}$Cs and $^{60}$Co sources. The obtained results were compared with the theoretical ones using a Phy-x/PSD software program version 0.1.0.0. It was found that the experimental and theoretical results are very agreed upon. Moreover, other nuclear radiation shielding parameters were evaluated. The results showed that the addition of bismuth oxide leads to an improvement in the ability of the composite glass to attenuate gamma rays by increasing the values of MAC and Z$_{eff}$, while it led to a decrease in the HVL and MFP, as well as the EBF and EABF. The results also showed that the addition of copper oxide led to an improvement in the ability of the present glass to slow down fast neutrons. Sample BiS40 showed the best result for gamma ray attenuation and sample BiS10 gave the best result for fast neutron removal cross section. The results were compared with some materials used for gamma ray shielding and fast neutron removal cross section, and it was concluded that samples Bi40 and BiS10 outperformed all commercial materials.

**Keywords:** radiation shielding; BiS40; borate glass; gamma rays; nuclear parameters

## 1. Introduction

The increasing use of radionuclides in nuclear reactors to obtain energy, as well as in treatment and medical imaging, and in laboratories for scientific research, requires more scientific studies to find suitable shields for gamma rays that are highly efficient in attenuating gamma rays and appropriate to transparency and thickness. The use of glass inlaid with heavy materials, such as bismuth and lead, can lead to a decrease in the thickness of the shielding material and reduce the level of gamma rays to a safe level, and the transparent glass material is better than other materials for shielding, such as conventional concrete used in nuclear reactors [1–3].

Among all types of glass, borate glass has properties that enable it to be used as a protective shield against gamma rays or neutrons because it is transparent to light in the visible region and can absorb a large and varied number of heavy elements, in addition to having thermal stability and a low melting point. Bismuth is distinguished from lead in that it is environmentally friendly, unlike lead, which has toxic effects on human health and is harmful to the environment. Therefore, researchers have recently focused on the use of bismuth as an additive in glass to be used as a protective shield for gamma rays with borate glass as an essential component due to its superior ability to slow down fast neutrons [4–7]. Borate glasses also feature different and complex structural units in their internal structure such as di, tri, tetra, and pentaborate. These changes are due to the effect

of modifiers in the glass system, such as CuO and $Li_2O$, and these modifiers also alter the optical and physical properties of the glass lattice [8,9].

Recently, many glasses doped with oxides of heavy metals have been prepared for use as shields against gamma rays. Saleh et al. [1] prepared bismuth borate glasses doped with lead and lithium, and the results showed that these prepared glasses have the best radiation protection compared with commercial glass as well as concrete used for protection from gamma rays. The only drawback of these glasses is that they contain lead, which is toxic to human health and the environment. Many researchers also prepared borate glass doped with bismuth or lead and other elements and it provided good results, but most of these works were theoretically studied by programs to determine their effectiveness in radiation protection [10–14].

When doped with oxides of heavy metals, such as lead and bismuth oxides, the glass exhibits a high atomic number and high density due to the high molecular mass of the heavy metals. These features make glass an effective material for absorbing harmful gamma rays. The environmental toxicity of lead (Pb) and its hazardous nature to human health are serious concerns among researchers for replacing lead with other high Z-materials such as Ba, Gd, W, and Bi, etc. Heavy metal oxide (HMO) glasses containing bismuth (Bi) or barium (Ba), with high refractive index and non-toxicity, show extremely high radioactive resistance due to their high atomic number and high density. [1,2].

In the presented work, copper–lithium-modified bismuth borate glasses were synthesized to provide radiation-protective, non-toxic glasses. The copper element was introduced for its effectiveness in absorbing fast neutrons, and lithium reduces the melting temperature of the sample. The effectiveness of the prepared samples for protection against gamma rays and neutrons was studied experimentally and theoretically by studying the parameters of the nuclear shielding and the effective removal cross section for fast neutrons. The non-crystalline structure of the samples was also confirmed by the XRD study.

## 2. Experimental Work
*Synthesis of Glass Samples*

The present glass samples have been prepared according to the chemical composition $10Li_2O + 20CuO + xBi_2O_3 + (70 - x)B_2O_3$, where x = 0, 10, 20, 30, 40 mol% from chemically pure substances $H_3BO_3$, $Bi_2O_3$, $CuCO_3$, $Li_2CO_3$ by the melt-quenching method. The materials used for synthesis glasses were weighed with a sensitive scale on a sensitivity of 0.0001, according to the previous equation, and the samples were mixed and crushed by mortar and pestle. The sample was placed in ceramic crucibles and placed in the oven at a temperature of 1000 °C for one hour. The sample was stirred every ten minutes to ensure homogeneity of the sample. The sample was poured into molds prepared for this purpose, made from stainless steel, and preheated to a temperature of 350 °C. Immediately after casting, the sample was placed in another oven at a temperature of 350 degrees for annealing for two hours to reduce the internal tension. Five synthesis samples were labeled as: BiS0.0, BiS10, BiS20, BiS30, and BiS40 according to the $Bi_2O_3$ content. Figure 1 shows the picture of the synthesis glasses.

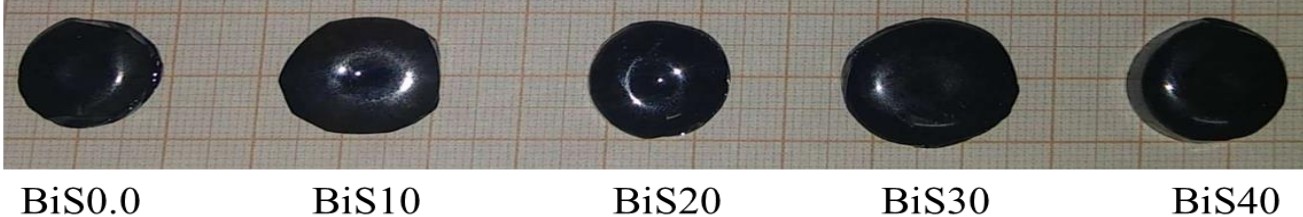

**Figure 1.** Glass system image.

### 3. X-ray Diffraction, Transmittance, and Density Measurements

To explore the crystallization of the synthesized glass samples, XRD assays were carried out using a Bruker D8 Discovered X-ray spectrometer, Billerica, Massachusetts, U.S and scanned with Cu Kα radiation (λ = 1.54 Å) at region 2θ 3–60°.

Optical absorption spectra were recorded using a Shimadzu ultraviolet spectrometer 2700 in the 200–800 nm spectral range.

Using the Archimedes principle of buoyancy, the density of the manufactured samples was measured at a room temperature of 25 °C. First, the samples were weighed in air ($w_a$) and then in toluene solution ($w_t$). The density of the manufactured samples was calculated using the following relationship:

$$\rho = \rho_t \frac{w_a}{w_a - w_t} \tag{1}$$

where $\rho_t$ is the toluene density (0.866 g·m⁻³).

*3.1. Gamma Ray Shielding Parameter Studies*

The material that should be used as a shield must be able to absorb or attenuate gamma rays. This can be estimated by the mass attenuation coefficient (μ/ρ). The values of the mass attenuation coefficient were determined experimentally for the prepared glass samples at the energy values of 662, 1173, and 1332 keV. For this purpose, narrow-beam transmission geometry was used. In this geometry, a parallel gamma ray beam reaches the detector passing through the prepared glass samples. Figure 2 illustrates the technique used in the gamma ray attenuation measurement. A $^{137}$Cs and $^{60}$Co sources were used for determining the mass attenuation coefficient experimentally with activities of 6.1 and 10 μCi, respectively. The spectrometer was calibrated for efficiency using a multi-nuclide standard solution (QCY 48) PT (Institute of radiology and radiation protection IRS, Hannover, Germany) (a mixed source containing $^{241}$Am, $^{57}$Co, $^{60}$Co, $^{85}$Sr, $^{88}$Y, $^{109}$Cd, $^{137}$Cs, $^{139}$Ce, and $^{203}$Hg.

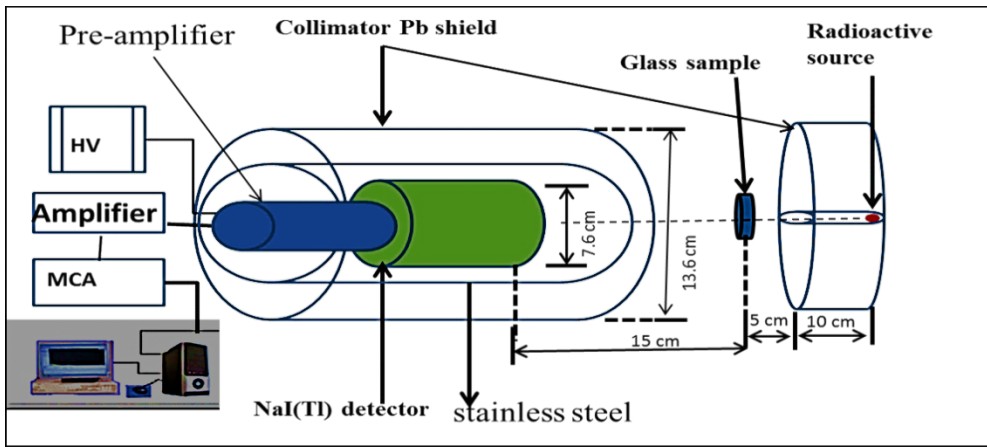

**Figure 2.** Experimental setup for gamma ray measurement.

The intensity of the incident ($I_0$) and transmitted (I) rays was measured by using Genie software, from Canberra, by placing the glass samples in the path between the source and the detector. The counting time was set to 10 min for each counting process, and it was measured three times for each sample and then the average was found. The background gamma intensity was also measured at the same time and subtracted from both readings. The mass attenuation coefficient was determined using the equation written as:

$$\frac{\mu}{\rho} = \frac{ln\frac{I_0}{I}}{\rho x} \tag{2}$$

where $I_0$ and I are the incident and transmitted intensities; $\rho$ is the density and x is the thickness of the glass samples (between 4 and 5.5 mm). The errors in calculating the mass attenuation coefficient are due to errors in the measurement of (1) the intensity of the incident and transmitted intensity, (2) the density, and (3) the thickness. The error is calculated by the error propagation formula given as:

$$\Delta\frac{\mu}{\rho} = \frac{1}{\rho x}\sqrt{\left(\frac{\Delta I_0}{I}\right)^2 (\Delta I/I)^2 + \left(ln\frac{I_0}{I}\right)^2 \left[\!\left[\left(\frac{\Delta\rho}{\rho}\right)^2 + \left(\frac{\Delta x}{x}\right)^2\right]\!\right]} \tag{3}$$

The mean free path (MFP) and half-value layer (HVL) were obtained from the following equations [15]:

$$MFP = \frac{1}{\mu} \tag{4}$$

$$HVL = \frac{ln2}{\mu} \tag{5}$$

### 3.2. Optimization of the Sample Thickness

The experimental aspect of this work was carried out using a single sample selected to meet Nordfor's criterion $2 < \mu t < 4$ [16]. At low energies and high Z-values, for which the thickness of a suitable sample does not exceed a few microns, the use of a single sample results in significant uncertainty in determining the thickness.

Optimization of the sample thickness—counting times was required to achieve 0.5% and 1% statistical accuracy. According to Nordfor's criterion, the counting rate required to achieve a statistical error rate of 0.5 and 1% is 0.5 s with an incoming flux of $10^6$ counts/s [17]. The use of this extended range criterion made it possible to use multiple samples covering a range of thicknesses greater than an order of magnitude.

In our current work, energies of 0.662, 1.173, and 1.332 MeV were used to find the mass attenuation coefficient, and the samples were cut to an appropriate thickness in the range of 4 to 5 mm, and the surfaces were finely polished Three samples of different thicknesses were prepared with an attenuation coefficient of $0.5 < \mu t < 6$ for each energy. Three measurements were taken for each sample at each photon energy.

### 3.3. Computational Work

Phy-x/PSD software program is an easy-to-use, online radiation shielding parameters measurement and dosimetry at "https://phy-x.net/PSD (accessed on 11 August 2022)" [18]. These parameters include all nuclear radiation parameters, such as LAC, MAC, MFP, HVL, TVL, $Z_{eff}$, $N_{eff}$, and energy exposure buildup factors (EABF) and exposure buildup factor (EBF). The program provides the values of the aforementioned factors in the energy from 1 keV to 100 GeV. In addition, by this program, it is possible to calculate the removal cross section for fast neutrons (FNRCS) of a compound or mixture.

## 4. Results and Discussion

### 4.1. XRD, Transmittance, Density, and Molar Volume

The XRD pattern for the present samples is illustrated in Figure 3. From the figure, it is clear that there are no sharp peaks in the spectra of the samples, which explains that the present samples are amorphous [18].

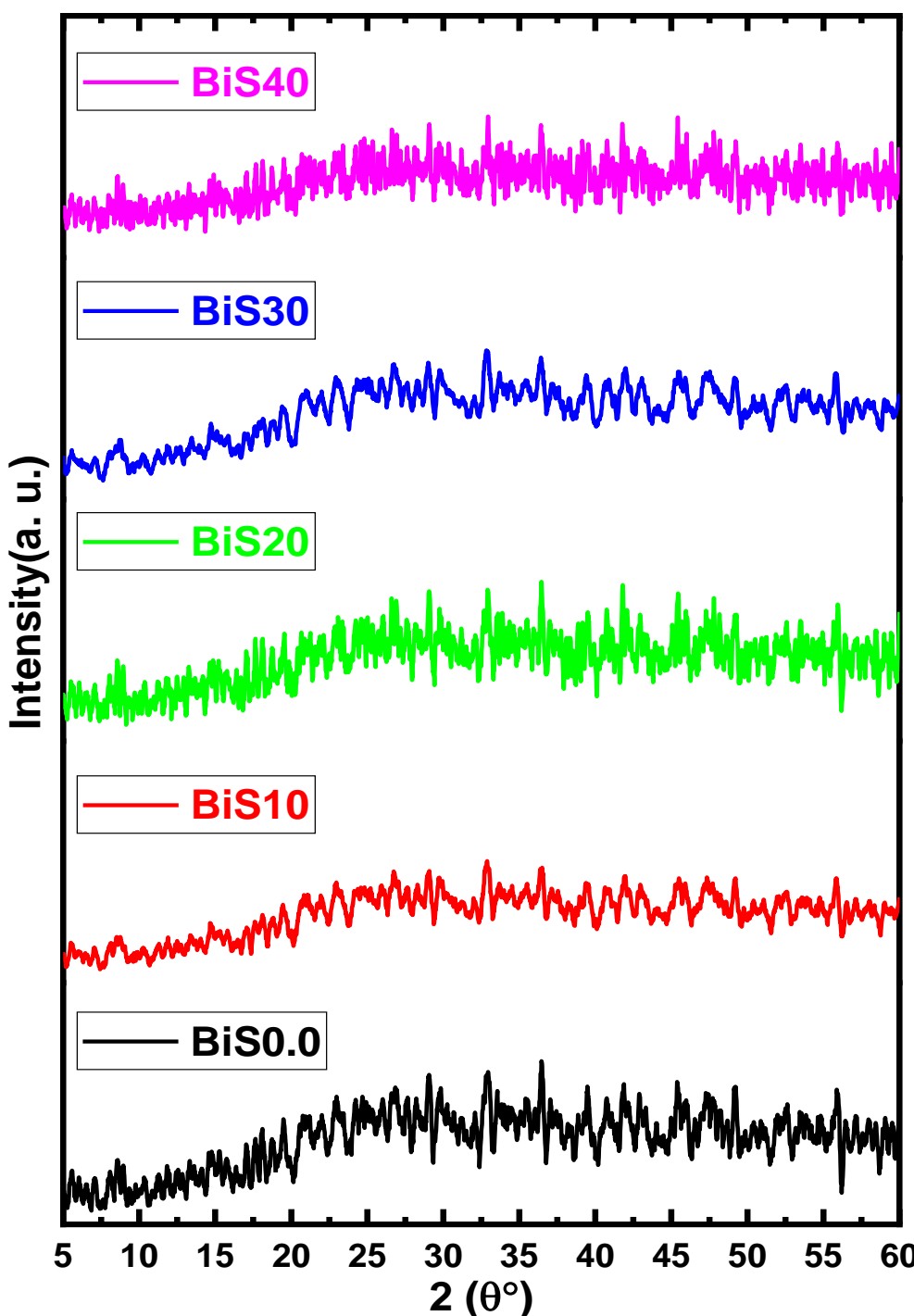

**Figure 3.** XRD patterns for the present samples.

The investigation of the basic absorption edge in the UV-VIS region is a particularly appealing approach for the physical transition of light via the band gap between the valance and conduction bands in both crystalline and amorphous media. The non-sharp edge in the absorption spectra of the glass samples indicates that the fabricated samples are amorphous. Figure 4 depicts the optical transmission spectra of all compositions. The optical absorption edge is not sharply defined, indicating the amorphous nature of the present glass system. It is also noticed that the cutoff wavelength increases as the content of $Bi_2O_3$ increases, and this indicates that electrons can easily move between the valence and conduction bands and that the defects increase with the increase in bismuth oxide content.

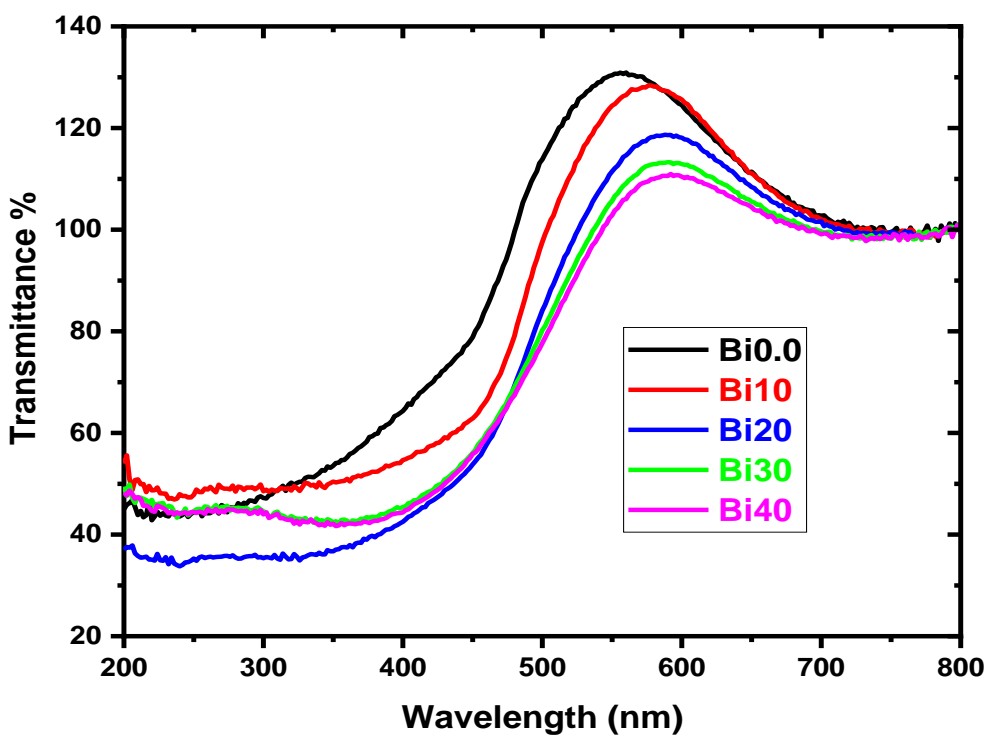

**Figure 4.** Transmittance of the glass samples versus wavelength for different $Bi_2O_3$ content.

Table 1 contains the density, molar volume, and chemical composition of the present glass samples. The density and molar volume are plotted in Figure 5. Density is one of the most important parameters that shows the structural change in the glass composition. As shown in Figure 4, the density rises from 2.813 to 5.584 $g \cdot cm^{-3}$ and the molar volume rises from 24.048 to 40.504 $cm^3 \cdot mol^{-1}$. The observed increase in density with increasing $Bi_2O_3$ content may be due to the molecular weight of $Bi_2O_3$ (465.96 $g \cdot mol^{-1}$) being larger than that of $B_2O_3$ (69.63 $g \cdot mol^{-1}$). In addition, the creation of non-bridging oxygen (NBO), which extends the structure of the loose network of the base glass system, may be responsible for the increase in density as the $Bi_2O_3$ concentration rises. Often, the density behavior is the opposite of that of the molar volume. However, in this study, the molar volume also increases with the increased content of $Bi_2O_3$, and this increase is due to the large ionic radius of bismuth ($Bi^{3+}$ = 1.03 Å) compared with the ionic radius of boron ($B^{3+}$ = 0.27 Å). This behavior appears in previous studies of similar types of glass under study [19,20].

**Table 1.** Chemical compositions, density, and molar volume of present glass samples.

| Sample Code | Mol% | | | | $\rho$ (g/cm³) | $V_m$ (cm³/mole) |
|---|---|---|---|---|---|---|
| | $B_2O_3$ | CuO | $Li_2O$ | BiO | | |
| BiS0.0 | 70 | 20 | 10 | 0.0 | 2.813 | 24.048 |
| BiS10 | 60 | 20 | 10 | 10 | 4.222 | 25.407 |
| BiS20 | 50 | 20 | 10 | 20 | 4.913 | 29.899 |
| BiS30 | 40 | 20 | 10 | 30 | 5.320 | 35.065 |
| BiS40 | 30 | 20 | 10 | 40 | 5.584 | 40.504 |

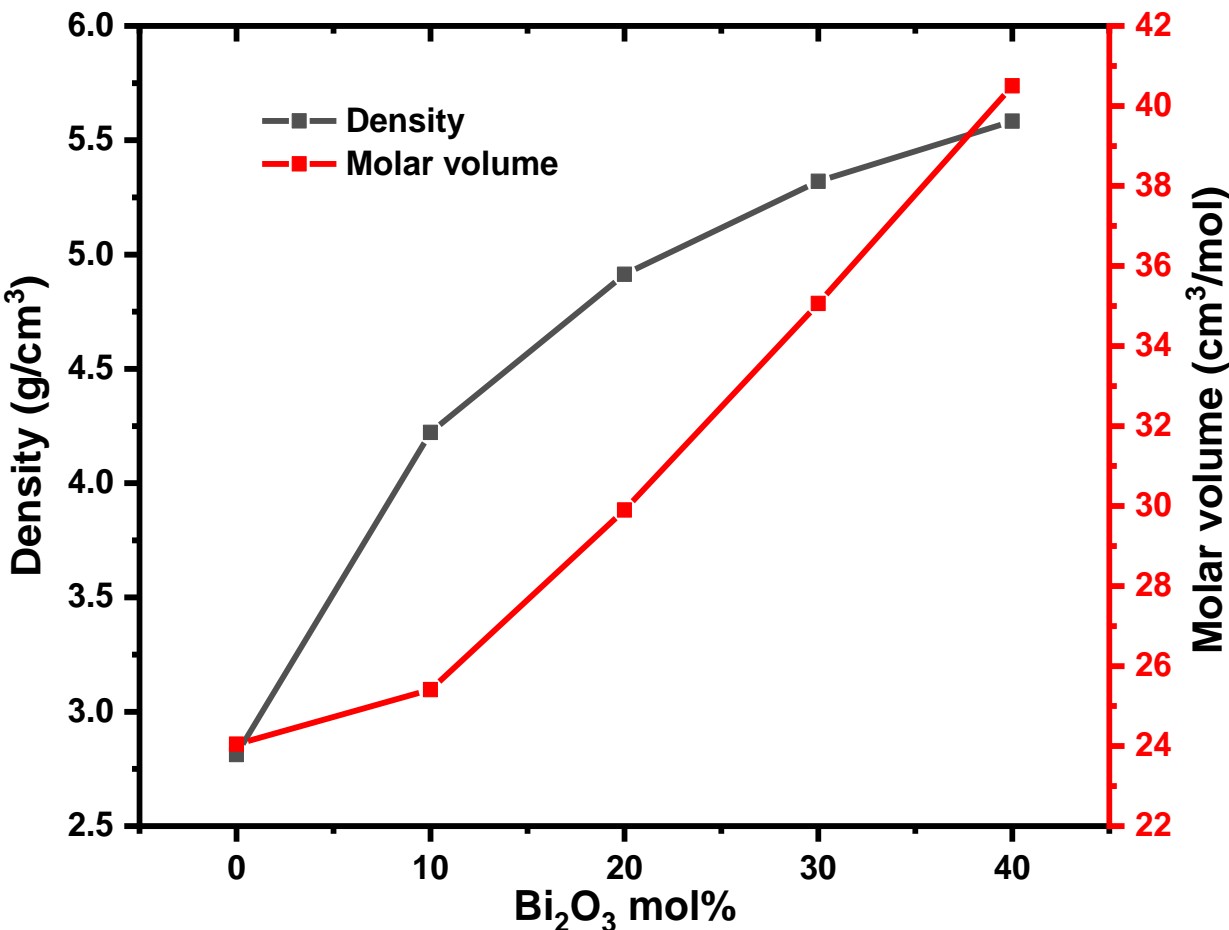

**Figure 5.** Density and molar volume for present glass samples.

*4.2. Nuclear Radiation Shielding*

4.2.1. Experimental Results

The experimental results of the MAC of the present samples are presented in Table 2 at energies 0.662, 1.173, and 1.333 MeV. The MAC values were calculated based on Equation (2). The results were compared with the theoretical results obtained by the Phy-x/PSD 0.1.0.0 software program. From the table, we notice that the experimentally calculated of MAC values agree with the values obtained by the Phy-x/PSD program, and the errors were within the limits of the experimental errors, which vary between 0.5% and 1.5%. Additionally, we noticed that the values of MAC increased with the increase in bismuth oxide in the samples. The highest value was for sample BiS40, which has more ability for radiation shielding, and the lowest value was for sample BiS0.0.

**Table 2.** MACs from experimental measurement and Phy-X/PSD software programs.

| Energy (MeV) | BiS0.0 | | BiS10 | | BiS20 | | BiS30 | | BiS40 | |
|---|---|---|---|---|---|---|---|---|---|---|
| | Exp. | Phy-X | Exp. | Phy-X | Exp. | Phy-X | Exp. | Phy-X | Exp. | Phy-X |
| 0.662 | $0.075 \pm 0.002$ | 0.075 | $0.089 \pm 0.003$ | 0.090 | $0.095 \pm 0.003$ | 0.096 | $0.100 \pm 0.003$ | 0.100 | $0.113 \pm 0.004$ | 0.103 |
| 1.173 | $0.056 \pm 0.002$ | 0.057 | $0.058 \pm 0.002$ | 0.059 | $0.059 \pm 0.002$ | 0.060 | $0.061 \pm 0.002$ | 0.061 | $0.060 \pm 0.002$ | 0.061 |
| 1.333 | $0.053 \pm 0.002$ | 0.053 | $0.054 \pm 0.002$ | 0.055 | $0.054 \pm 0.002$ | 0.055 | $0.054 \pm 0.002$ | 0.056 | $0.055 \pm 0.002$ | 0.056 |

4.2.2. Theoretical Results

In this work, the nuclear shielding parameters of the prepared samples BiS0.0–BiS40 were calculated using Phy-x/PSD software in the photon energy (PE) range from 1.5 to

15 MeV. The values of the mass attenuation coefficient versus the photon energy are plotted in Figure 6. The MAC values were found to be 15.07, 54.68, 72.91, 83.39, and 90.21 cm$^2$/g at 15 keV for BiS0.0, BiS10, BiS20, BiS30, and BiS40 samples, respectively. From the figure, we notice that the values of the MAC decrease dramatically as the photon energy increases, followed by a sudden increase at the energy of 90.5 keV, which is the absorption K-edge for bismuth, and then decrease until it reaches the energy value of 3 MeV. After 3 MeV energy, the MAC values begin to increase slowly in the middle energy of the photon. We also notice from the figure that the MAC values increase with the increase in the percentage of bismuth oxide in the samples. The sample BiS40 provided the highest values, and the lowest values were for the sample BiS0.0. For processes of photon interaction with matter and at lower energy values, the photoelectric reaction of the photon is dominant, and with increasing energy, that is, at average photon energies of 1–3 MeV, Compton scattering becomes dominant at this energy, and with an increase of PE above 3 MeV pair production is the predominant process.

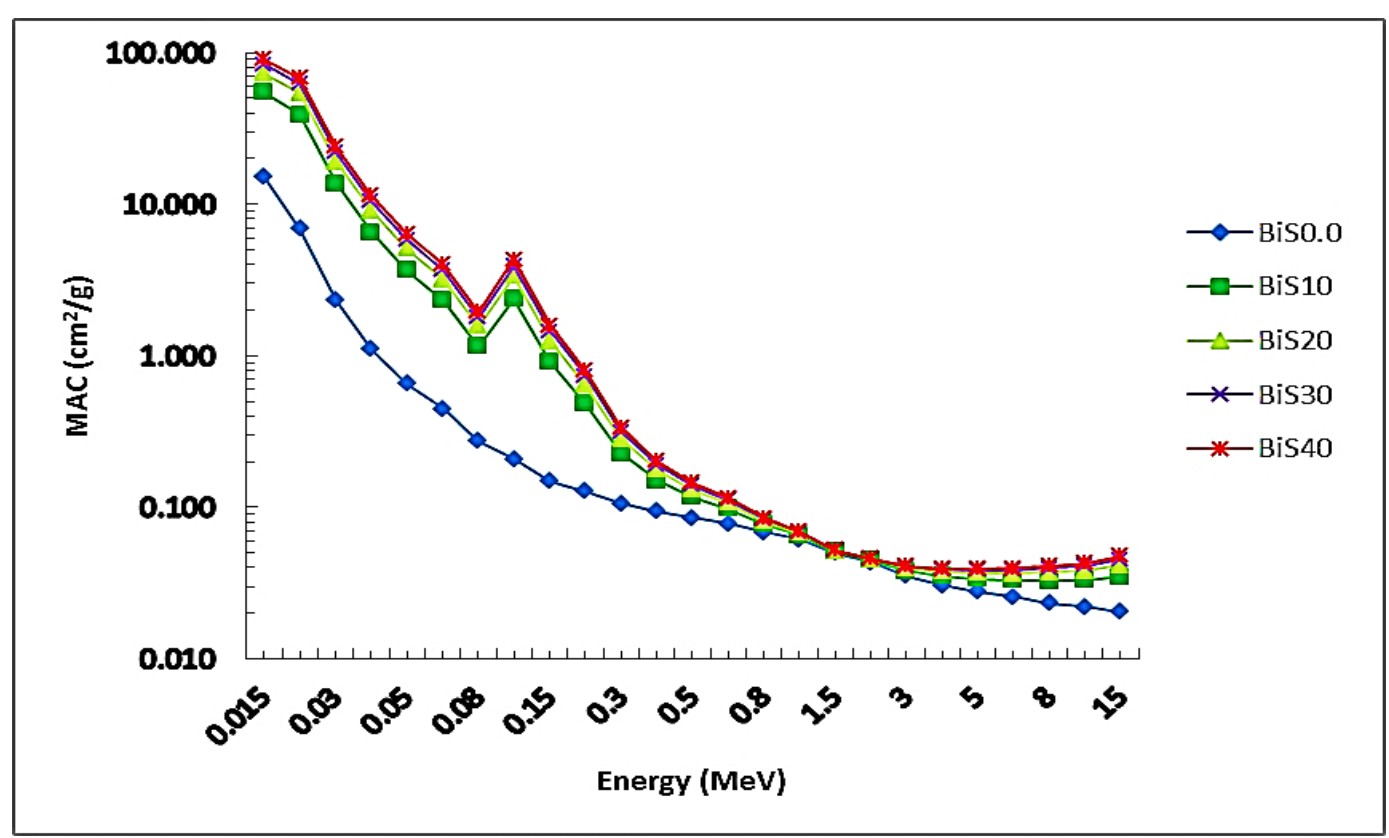

**Figure 6.** MAC values depending on the photon energy for all glass samples.

To study the effectiveness of the prepared samples against gamma rays and the extent of their prevention, the MAC values of the current sample BiS40 were compared with the MAC values of other commercial materials used for gamma radiation protection, such as barite concrete and ordinary concrete (BaC and OC), ferrite and chromite (Fe$_2$O$_3$ and FeCr$_2$O$_4$), and some types of glass (RS 520 G18 and RS 360) [21]. The results are plotted in Figure 7. It is clear from the figure that the results of sample BiS40 possess the highest values for the MAC in comparison with the previously mentioned materials, and, therefore, sample BiS40 can be used as an alternative to the previous materials for radiation protection.

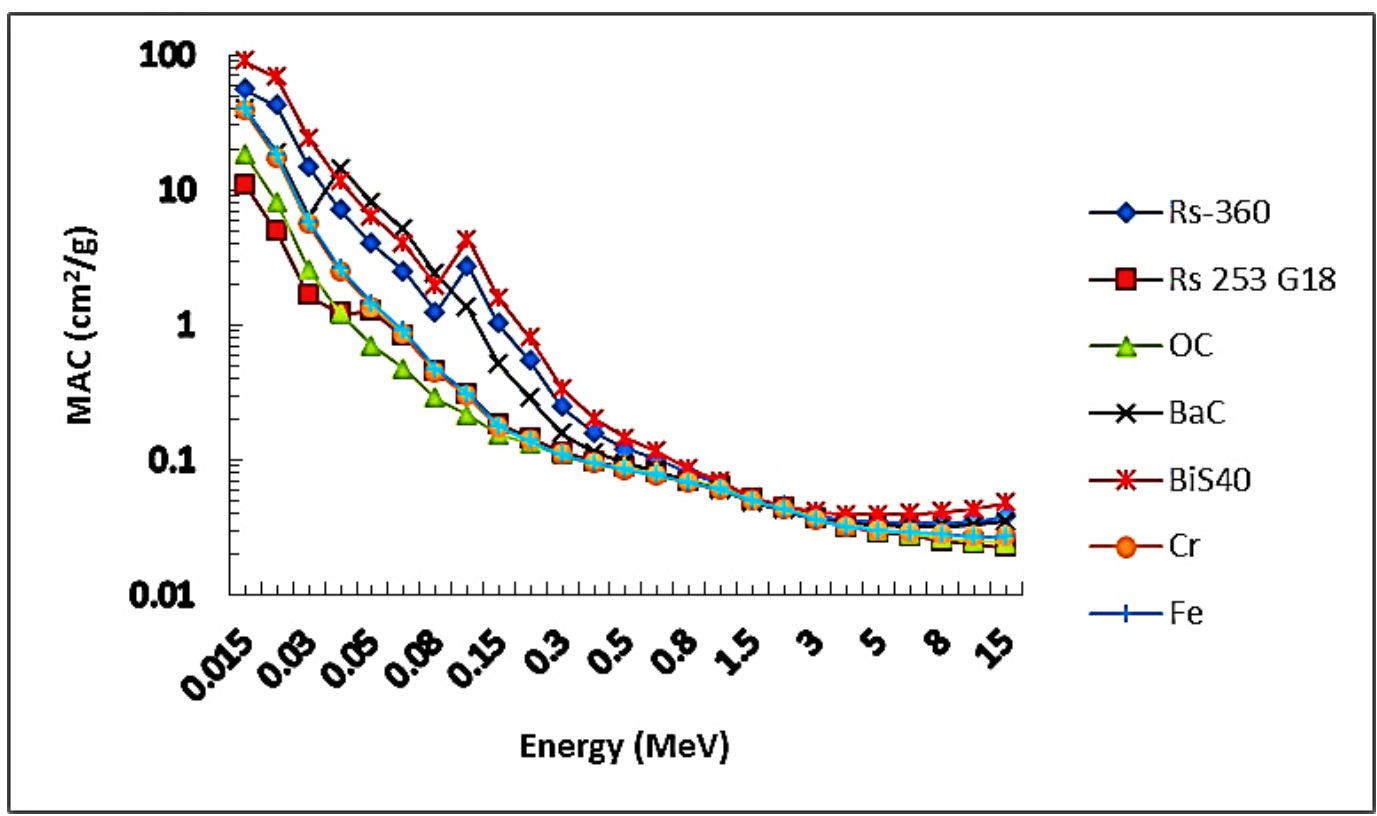

**Figure 7.** Comparison values of the MAC of the present sample BiS40 with commercial materials.

The HVL is defined as the material needed to reduce the intensity of incident photons to half their original value. The smaller size of the glass sample and the greater radiation blocking efficiency means better shielding applications. In addition, MFP (the distance a photon travels in a substance before another reaction occurs) is one of the main parameters for estimating the shielding properties of any material.

The values of HVL for the present samples were set versus the photon energy in Figure 8. From the figure, the lowest values were at 15 kV, and found to be 0.0169, 0.0030, 0.0019, 0.0016, and 0.0014 cm for samples BiS0.0, BiS10, BiS20, BiS30, and BiS40, respectively. The lowest values were for sample BiS40, and the largest values were for sample BiS0.0. After 15 keV of photon energy, the HVL values start to gradually rise as the photon energy increases. Then, it increases slowly before the photon's energy reaches 15 MeV. The results for HVL values for sample BiS40 were compared with the HVL for concrete and commercial glass used as a shield to gamma rays, and Figure 8 shows the results. It is clear from Figure 9, that the values of the HVL of the prepared sample were lower than those of the commercial materials mentioned previously, and, therefore, the present sample of the prepared glass can be used as a substitute for the previous commercial materials in radiation protection applications.

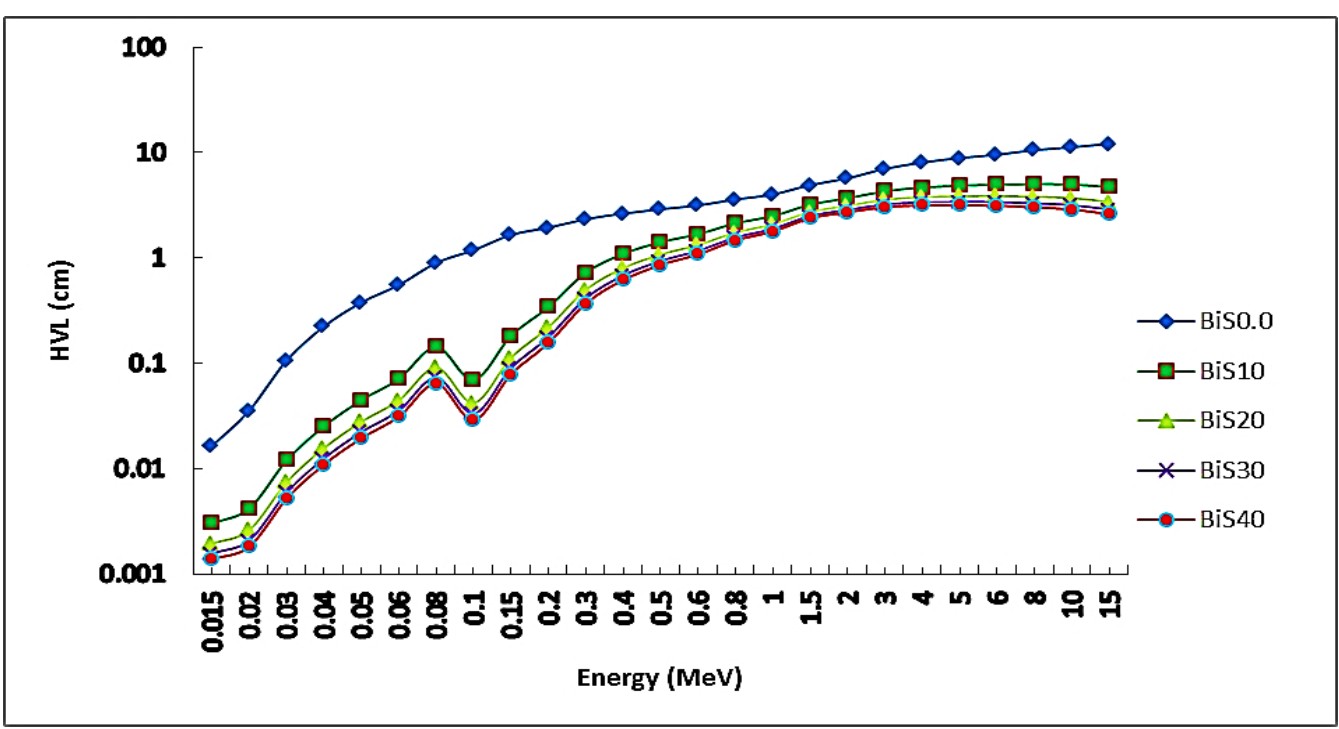

**Figure 8.** HVL depending on photon energy for all present glass samples.

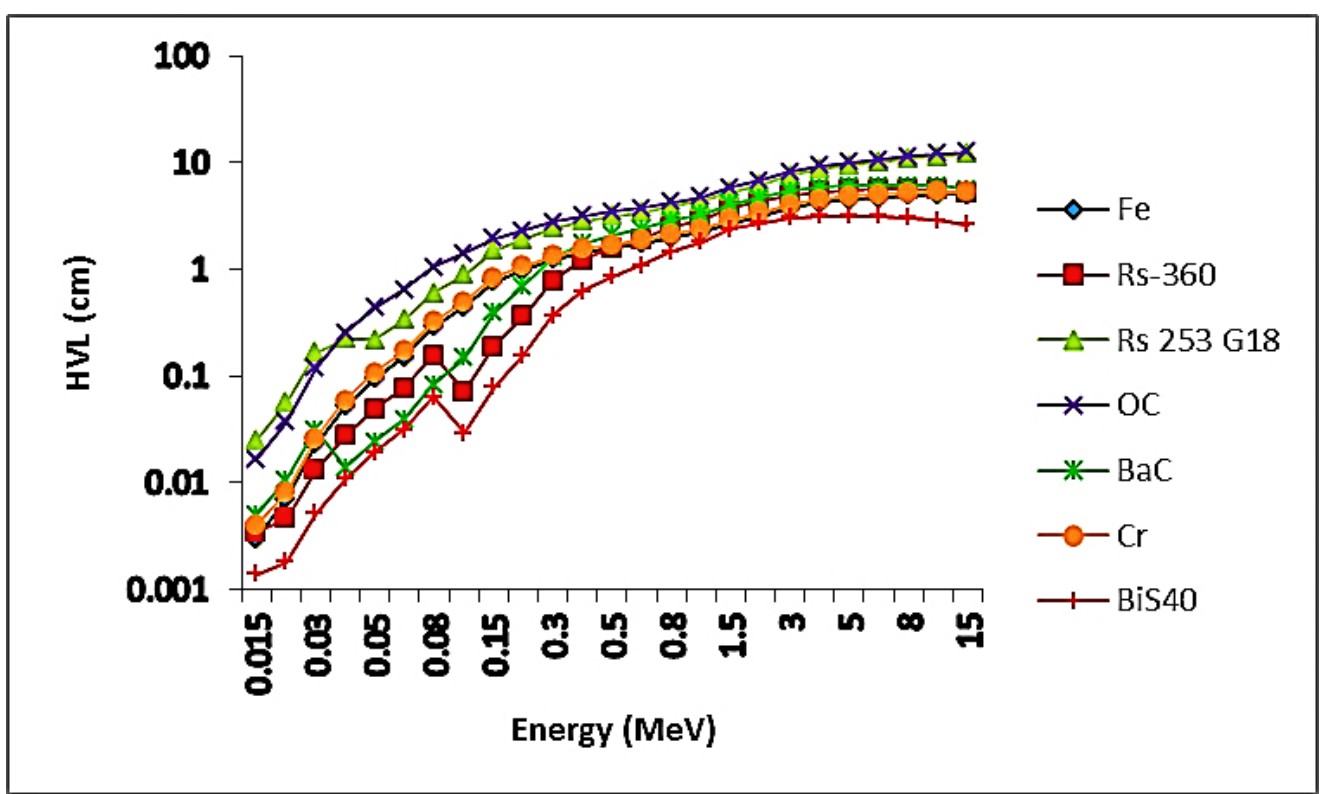

**Figure 9.** Comparison values of the HVL of the present sample BiS40 with commercial materials.

Figure 10 shows the MFP versus photon energy of the present samples. From the figure, at all values of the photon energies, we can noticed that the values are affected by the addition of $Bi_2O_3$ to the prepared glass system due to the increased density of the prepared

glass samples. Additionally, the values of MFP increase steadily until the photon's energy runs to 3 MeV. After 3 MeV values of photon energy, the MFP values slightly increase until the photon's energy reaches the value of 15 MeV. At a photon energy of 15 keV, the values of MFP were approximately 0.024, 0.004, 0.003, 0.0023, and 0.0019 for samples BiS0.0–BiS40, respectively. The best results were for sample BiS40, due to the high content of bismuth oxide and the increase in the density of the sample.

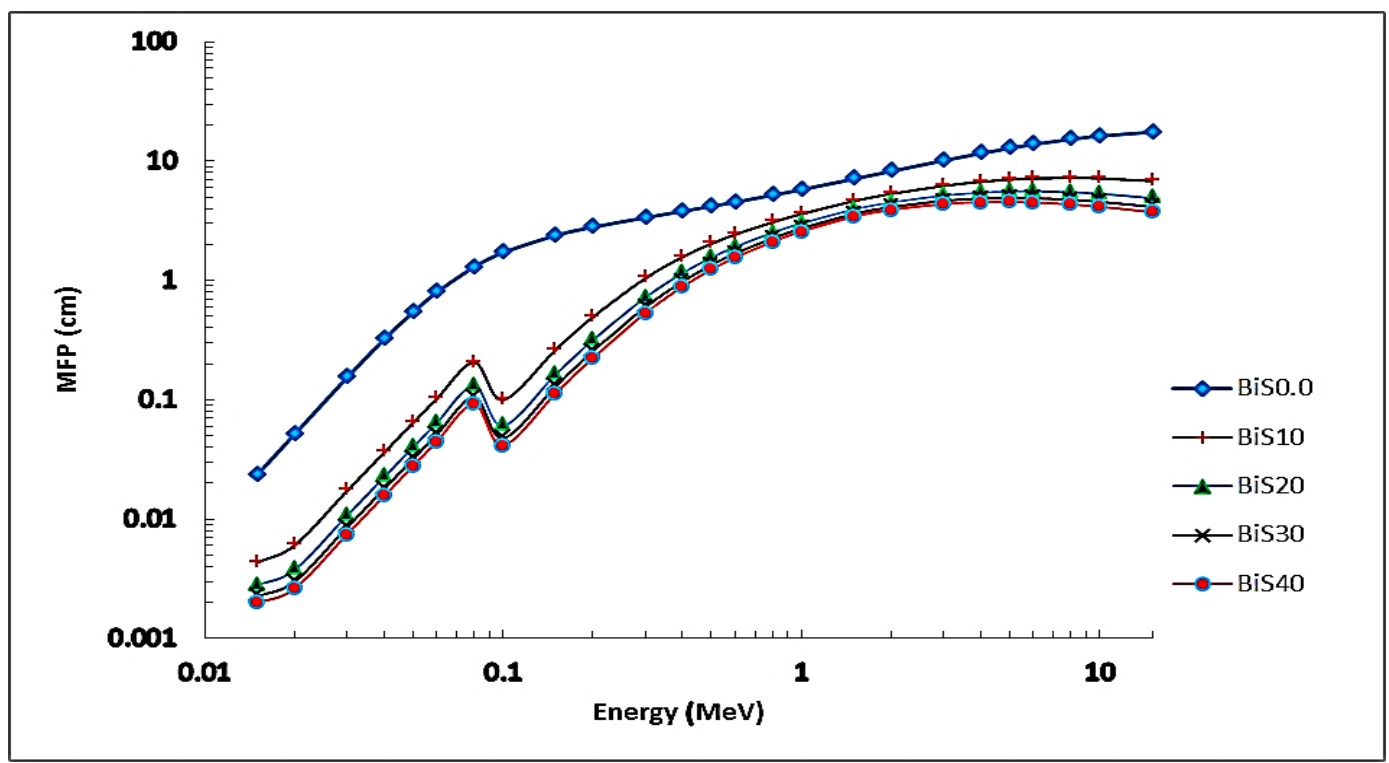

**Figure 10.** MFP depending on photon energy for all present glass samples.

Figure 11 shows the values of the $Z_{eff}$ with the photon energy. It was noticed that the prepared glass samples had an increase in the $Z_{eff}$ values with the increase in the proportion of $Bi_2O_3$ in the samples. In addition, the maximum $Z_{eff}$ values were observed to be in the photon energy range from 0.015 to 0.01 MeV. After that, from 0.01 to 2 MeV, the $Z_{eff}$ values decrease rapidly as the energy of the incident photon increases. From 2 to 15 MeV for photon energy, the values of the effective atomic number increase significantly, due to the pair production reaction, which is dominant in this region.

The values of the effective atomic number of the samples were found to be 23.8, 58.1, 67.3, 71.6, and 74.0 for BiS0.0–BiS40 samples, respectively, and the best values were for sample BiS40 because it contained a high percentage of bismuth oxide, which led to an increase in the density of the sample. In comparison with the other shielding systems, the results are shown in Figure 12. It can be noticed from the figure, that the prepared sample BiS40 obtained the best results for the $Z_{eff}$, and, therefore, it is considered the best shielding system.

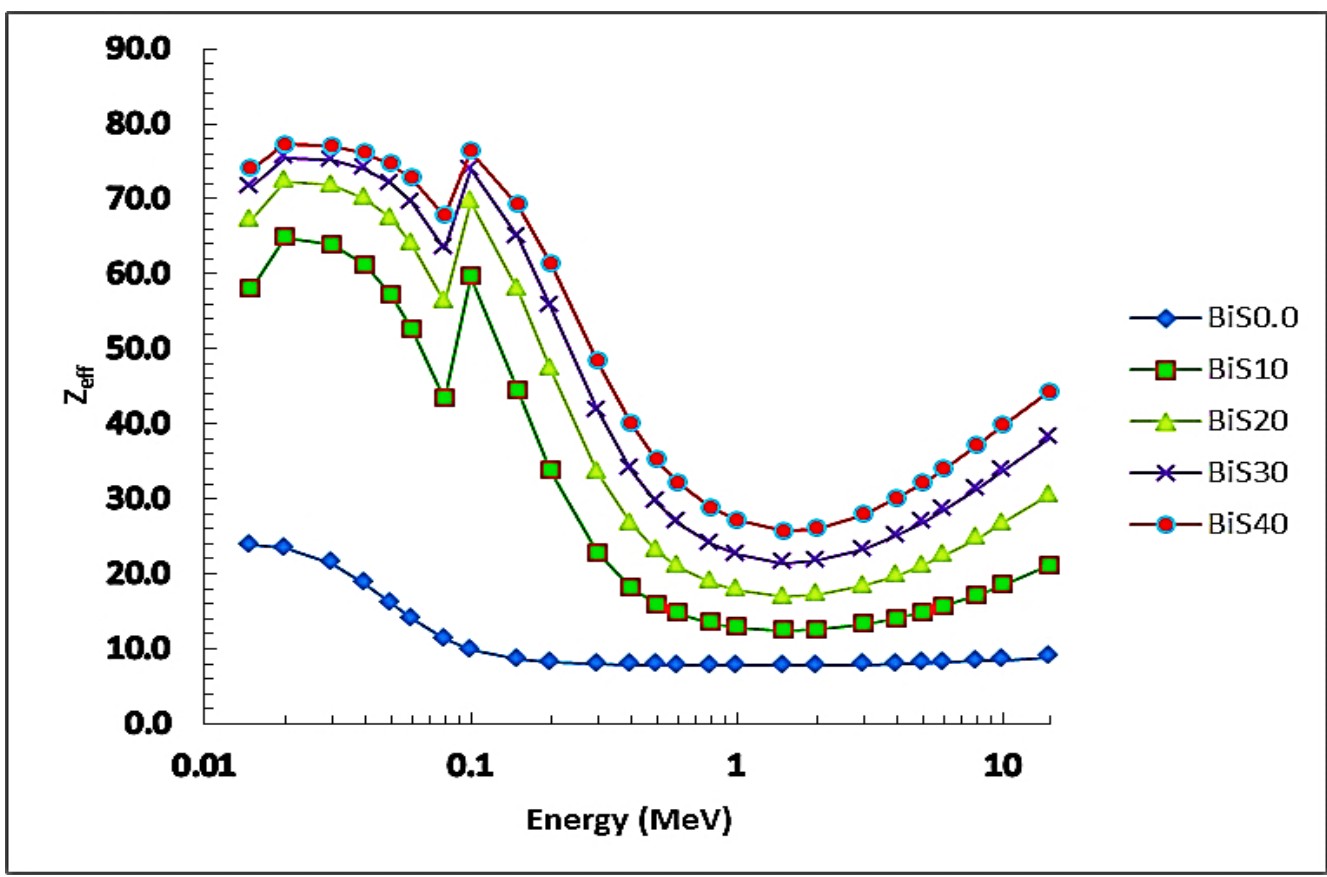

**Figure 11.** $Z_{eff}$ depending on photon energy for all present glass samples.

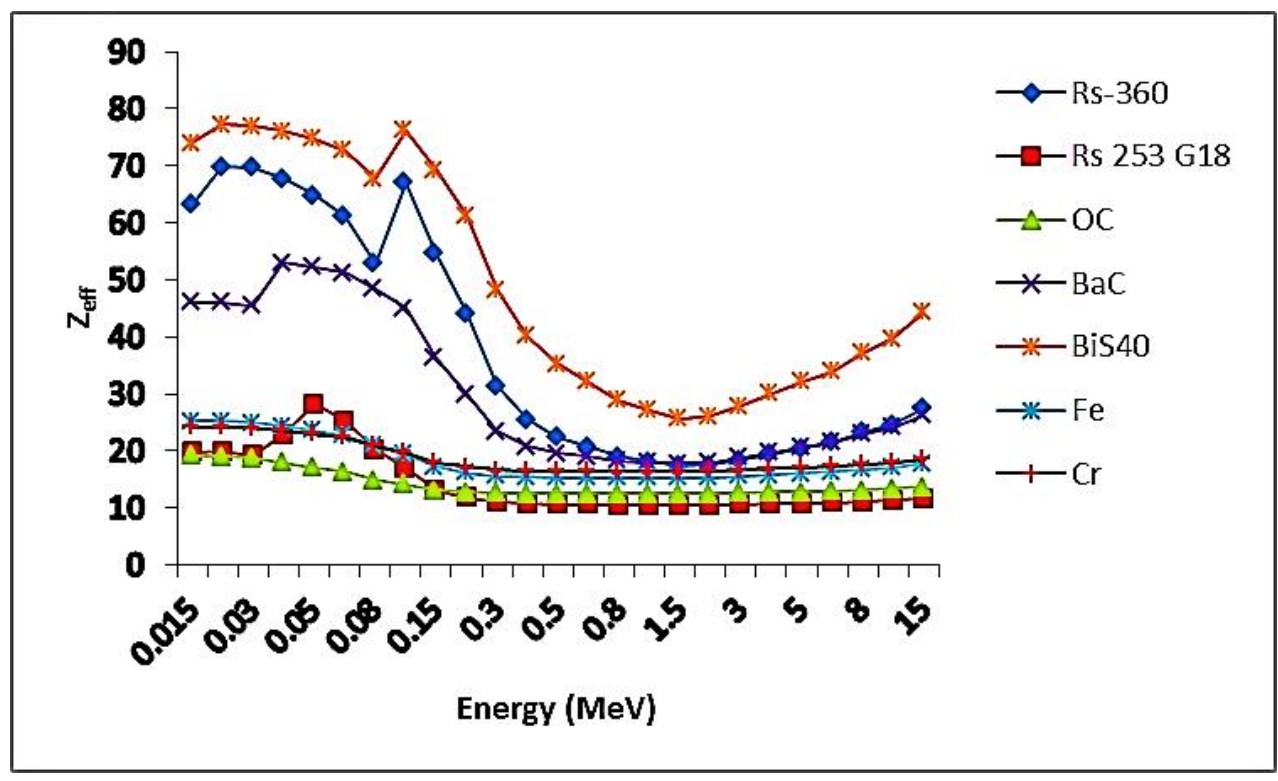

**Figure 12.** Comparison values of the $Z_{eff}$ of the present sample BiS40 with commercial materials.

Figures 13 and 14 show EBF and EABF values of the present glass system with a penetration depth of 1, 10, 20, 30, and 40 MFP at a photon energy values range 0.015–15 MeV. In general, it was observed that EBF values increase with the increase in photon energy and penetration depth. Concerning the sample BiS0.0, which is free of $Bi_2O_3$, from the figure, it can be seen that the EBF values begin to increase with the increase in the energy of the photon, until it reaches the highest value at energy 0.3 MeV, and then gradually decrease with increasing energy. It was also noticed from the figure that two sharp peaks appeared dramatically for the BiS10–BiS40 samples at the photon energy values 20 and 90 keV due to the absorption L and K-edge of bismuth.

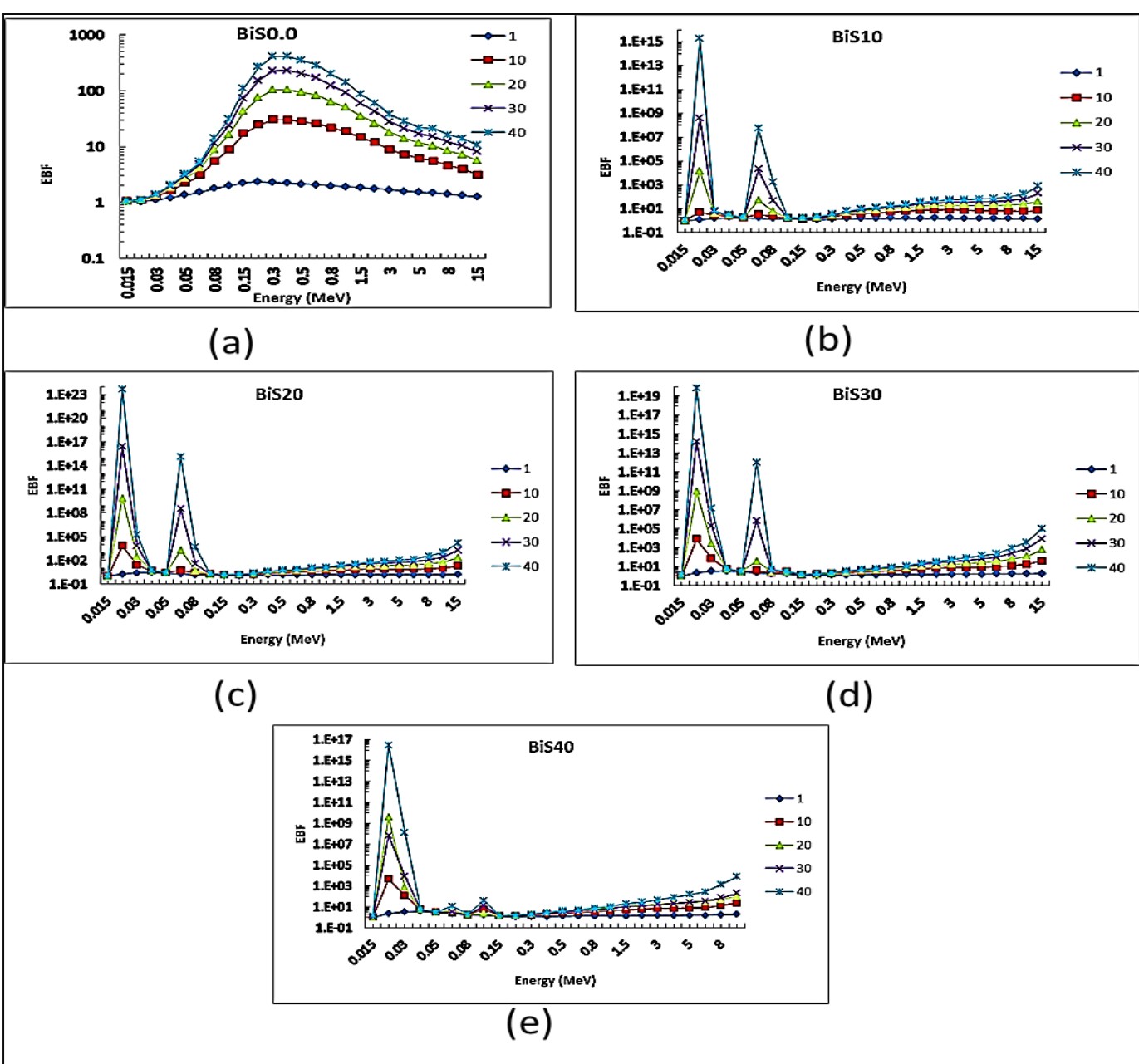

**Figure 13.** EBF depending on photon energy for all present glass samples: (**a**) Bi0.0, (**b**) Bi10, (**c**) Bi20, (**d**) Bi30, (**e**) Bi40.

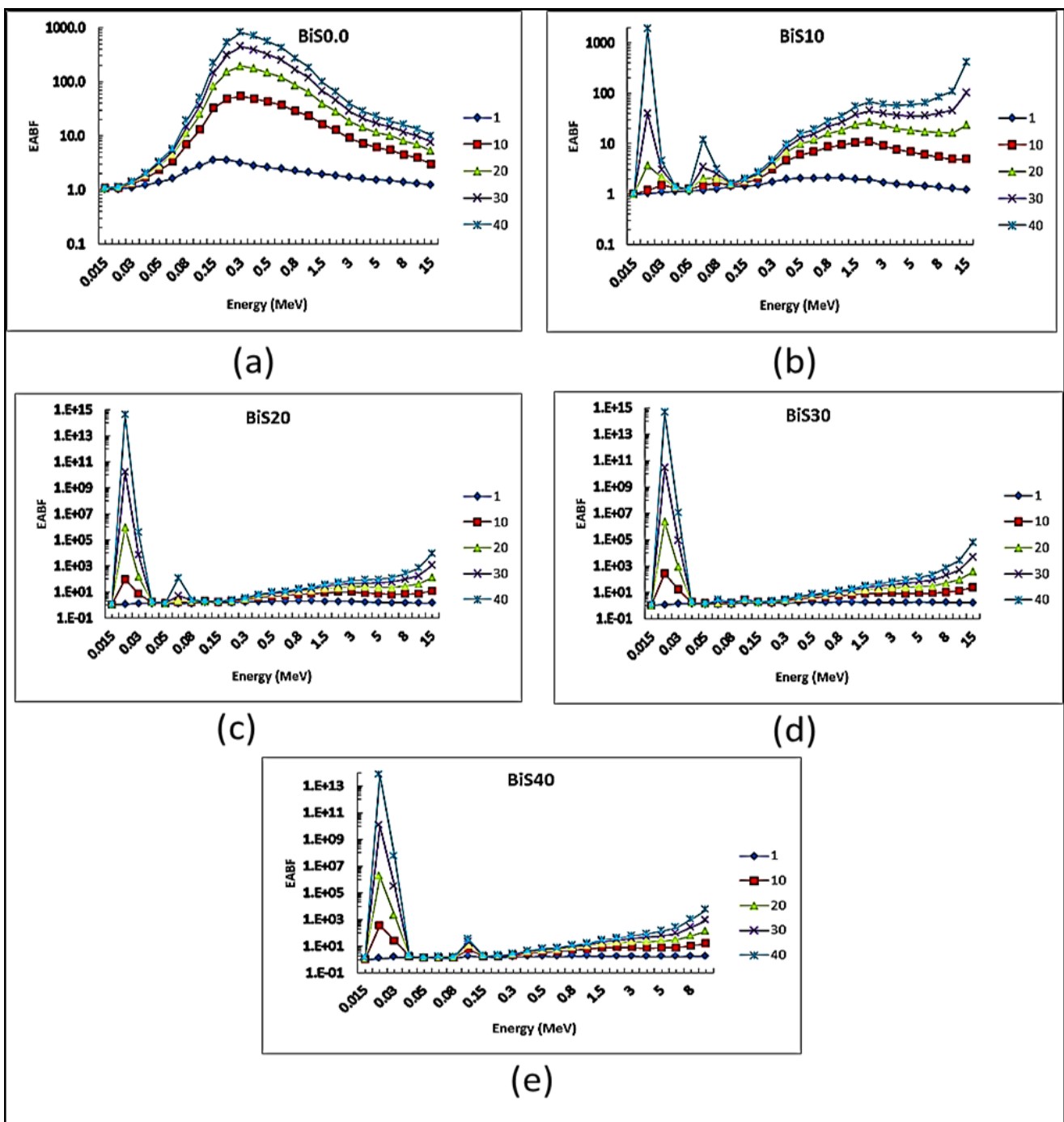

**Figure 14.** EABF depending on photon energy for all present glass samples: (**a**) Bi0.0, (**b**) Bi10, (**c**) Bi20, (**d**) Bi30, (**e**) Bi40.

There are differences in the values of EBF of the studied samples and this difference is due to the different concentrations of bismuth oxide in the glass system.

The fast neutron removal cross section ($\Sigma_R$) was also found for all studied present samples. Figure 15 shows the values of $\Sigma_R$ for all present samples. From the figure, the values of $\Sigma_R$ for the present samples were 0.116, 0.121, 0.113, 0.105, and 0.099 cm$^{-1}$ for BiS0.0–BiS40, respectively. The best value of the $\Sigma_R$ was 0.121, which is for the sample BiS10; after that, a decrease in the value of $\Sigma_R$ can be noticed with the increase in the bismuth content in the samples. The reason for this decrease is due to the decrease in the boron content of the samples at the expense of the increase in the bismuth content. By comparing

the $\Sigma_R$ value of sample BiS10, it was observed that it outperformed most of the materials used in standard shielding, such as graphite = 0.077 cm$^{-1}$, ordinary concrete = 0.094 cm$^{-1}$, and water = 0.1023 cm$^{-1}$.

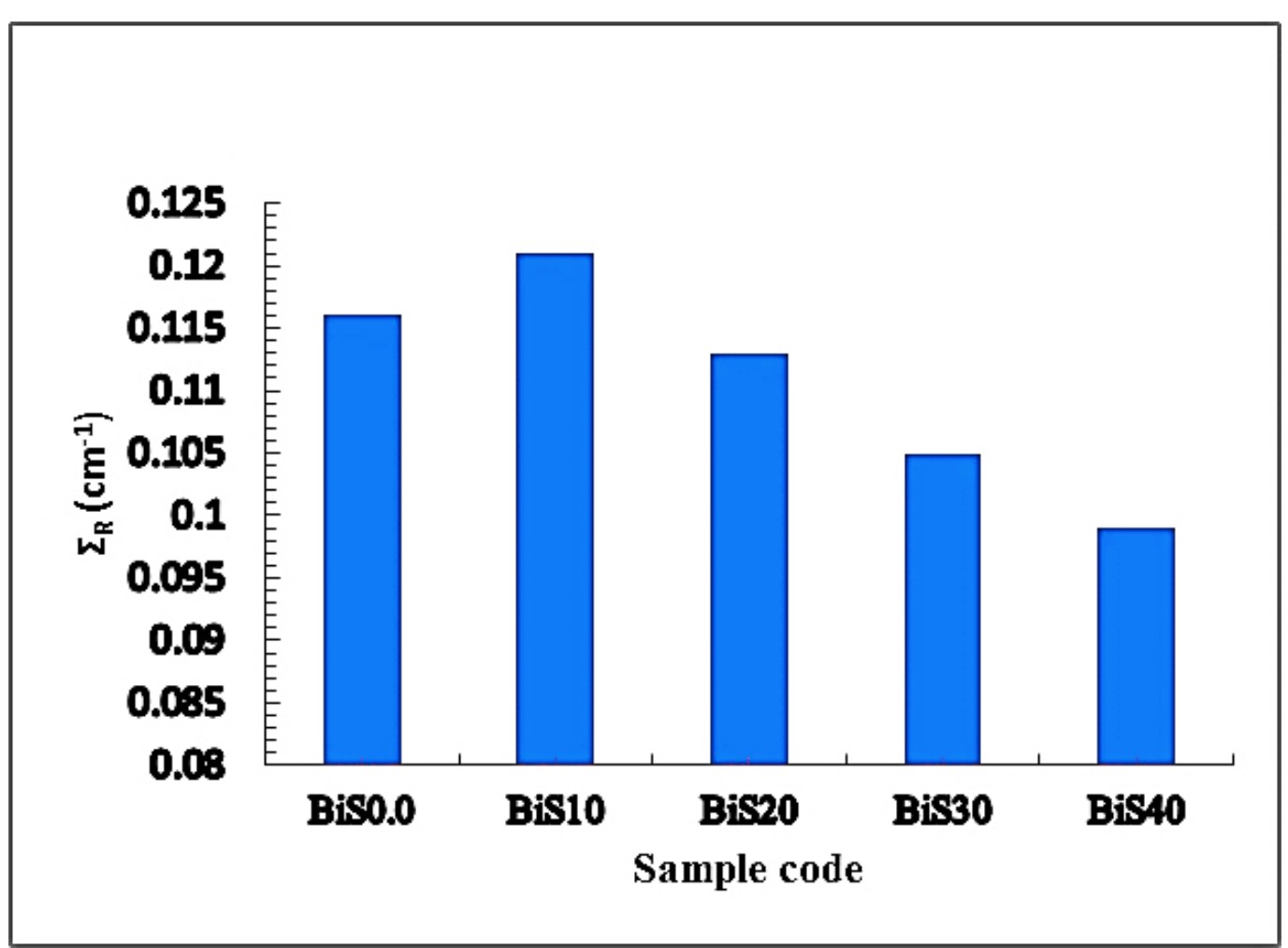

**Figure 15.** $\Sigma_R$ for all present glass samples.

### 5. Conclusions

The physical properties, as well as the shielding properties, of gamma rays and neutrons were experimentally and theoretically investigated for the $10Li_2O + 20CuO + xBi_2O_3 + (70 - x)B_2O_3$, where x = 0, 10, 20, 30, 40 mol% glass matrix. It was found that the density of samples increased from 2.813 to 5.584 (g/cm$^3$), and Vm increased from 24.155 to 27.074 (cm$^3$/mol). XRD examination confirmed the non-crystalline character of the samples under investigation. The results showed that MAC and $Z_{eff}$ of the current glass samples increased with increasing $Bi_2O_3$ content, and both the HVL and MFP decreased with increasing the content of $Bi_2O_3$. By comparing the values of MAC, $Z_{eff}$, and HVL with many materials used in gamma ray shields, it was found that the samples under study outperformed all aforementioned commercial materials used as shields for gamma rays. The cross section of the effective removal of fast neutrons was also studied, and it was found that it is higher than the cross section of the effective removal of water, paraffin, graphite, and concrete.

Therefore, the experimental and theoretical study proved that sample BiS40 has much better gamma ray shielding properties than other materials used as gamma ray shields.

**Author Contributions:** Conceptualization, E.E.S. and M.S.A.-F.; methodology, E.E.S.; software, E.E.S.; validation, M.S.A.-F. and F.A.; formal analysis, F.A.; investigation, E.E.S. and M.S.A.-F.; resources, M.S.A.-F. and F.A.; data curation, F.A.; writing—original draft preparation, E.E.S. and M.S.A.-F.; writing—review and editing, E.E.S. and M.S.A.-F.; visualization, E.E.S. and F.A.; project administration, M.S.A.-F.; funding acquisition, M.S.A.-F. and F.A. All authors have read and agreed to the published version of the manuscript.

**Funding:** This research was funded by [The Deputyship for research and innovation, Ministry of Education, Saudi Arabia and Qassim university] grant number [QU-IF-2-5-3-26870].

**Acknowledgments:** The authors extend their appreciation to the Deputyship for Research and Innovation, Ministry of Education, Saudi Arabia for funding this research work through the project number (QU-IF-2-5-3-26870). The authors also thank to Qassim University for technical support.

**Conflicts of Interest:** The authors declare no conflict of interest.

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
