# Peer review of "Synthesis of Novel Li2O-CuO-Bi2O3-B2O3 Glasses for Radiation Protection: An Experimental and Theoretical Study"

_inorganics, doi:10.3390/inorganics11010027_

Round 1

Reviewer 1 Report

Authors study the borate glass samples with varying composition and their ability to protect from the gamma-radiation. The source of radiation were 137Cs and 60Co sources. The article claims to be both experimental and theoretical though most of the results are obtained by Phy-X/PSD software available online as a web-service. This software is described by its authors in the reference [15] which would better be repeated near the link to the software itself in section 2.3.

The manuscript is close by its structure to the recent paper of authors (Journal of Electronic Materials (2022) 51:7283–7296) on different compound of Li2O‑ZnO‑P2O5 family. Most of the material is in fact the visualization of the Phy-X/PSD results that can be easily exported to the MS Excel format. However in the latter article (not referenced in the manuscript) the authors put more effort to the computational method description and figures presentation.Thus I have the following comments to be addressed before possible publication:

1. Excel-style plots with bounding boxes could be restyled to look more professionally. Figures with multiple frames to be named a), b), ... and better described in figure captions. In case of sharp peaks in Fig. 12 it isn't physically good to connect dots with lines, as real peaks should be much sharper than the ones drawn by lines.

2. From the experimental point of view, did the authors measure optical transmission of samples under study? They all seem opaque from the picture. Is it not important for simple usage as protective layer?

3. Authors should better describe the preparation procedure for samples before measurements. Were they cut and polished? Though the optical-grade surface quality may not be required for gamma measurements, the accuracy of thickness determination depends on the surface type.

4. In section 2.2 the flux is stated to be 106, where it is 10^6 counts/s in the reference 17.

5. XRD results in Fig. 3 all show an amorphous halo, which is expected for glass samples. To me, the Fig. 3 may be omitted and replaced by a statement in the text.

6. Values in Table 2 may be rounded to the third digit as the uncertainty is in the order of 0.001 - 0.003.

7. I would add some extra references to the introduction, for instance  P. Kaur et al. / Spectrochimica Acta Part A 206 (2019) 367–377 or . Limkitjaroenporn et al. / Journal of Physics and Chemistry of Solids 72 (2011) 245–25.

Author Response

Comments and Suggestions for Authors

Authors study the borate glass samples with varying composition and their ability to protect from the gamma-radiation. The source of radiation were 137Cs and 60Co sources. The article claims to be both experimental and theoretical though most of the results are obtained by Phy-X/PSD software available online as a web-service. This software is described by its authors in the reference [15] which would better be repeated near the link to the software itself in section 2.3.

The manuscript is close by its structure to the recent paper of authors (Journal of Electronic Materials (2022) 51:7283–7296) on different compound of Li2O‑ZnO‑P2O5 family. Most of the material is in fact the visualization of the Phy-X/PSD results that can be easily exported to the MS Excel format. However in the latter article (not referenced in the manuscript) the authors put more effort to the computational method description and figures presentation.Thus I have the following comments to be addressed before possible publication:

  1. Excel-style plots with bounding boxes could be restyled to look more professionally. Figures with multiple frames to be named a), b), ... and better described in figure captions. In case of sharp peaks in Fig. 12 it isn't physically good to connect dots with lines, as real peaks should be much sharper than the ones drawn by lines.
  • Figures 12 and 13 have been reformulated with multiple frames, according to your suggestion.
  1. From the experimental point of view, did the authors measure optical transmission of samples under study? They all seem opaque from the picture. Is it not important for simple usage as protective layer?
  • The transmittance of the studied samples were measured and added to the experimental part and the results. Figure 4 has been added.
  1. Authors should better describe the preparation procedure for samples before measurements. Were they cut and polished? Though the optical-grade surface quality may not be required for gamma measurements, the accuracy of thickness determination depends on the surface type.
  • The preparation of samples for gamma measurements was explained in the section of gamma measurements. The samples were cut and polished before the measurement process, with thicknesses ranging from 4 to 5 mm.
  1. In section 2.2 the flux is stated to be 106, where it is 10^6 counts/s in the reference 17.
  • Done
  1. XRD results in Fig. 3 all show an amorphous halo, which is expected for glass samples. To me, the Fig. 3 may be omitted and replaced by a statement in the text.
  • XRD tests are necessary to show that the prepared glass is amorphous. This procedure requires showing the results of these tests through Figure 2.
  1. Values in Table 2 may be rounded to the third digit as the uncertainty is in the order of 0.001 - 0.003.
  • The values in Table 2 have been revised as your suggestion.
  1. I would add some extra references to the introduction, for instance P. Kaur et al. / Spectrochimica Acta Part A 206 (2019) 367–377 or . Limkitjaroenporn et al. / Journal of Physics and Chemistry of Solids 72 (2011) 245–25.
  • Your suggested reference has been added.

Thank you for reviewer.

Comments and Suggestions for Authors

Authors study the borate glass samples with varying composition and their ability to protect from the gamma-radiation. The source of radiation were 137Cs and 60Co sources. The article claims to be both experimental and theoretical though most of the results are obtained by Phy-X/PSD software available online as a web-service. This software is described by its authors in the reference [15] which would better be repeated near the link to the software itself in section 2.3.

The manuscript is close by its structure to the recent paper of authors (Journal of Electronic Materials (2022) 51:7283–7296) on different compound of Li2O‑ZnO‑P2O5 family. Most of the material is in fact the visualization of the Phy-X/PSD results that can be easily exported to the MS Excel format. However in the latter article (not referenced in the manuscript) the authors put more effort to the computational method description and figures presentation.Thus I have the following comments to be addressed before possible publication:

  1. Excel-style plots with bounding boxes could be restyled to look more professionally. Figures with multiple frames to be named a), b), ... and better described in figure captions. In case of sharp peaks in Fig. 12 it isn't physically good to connect dots with lines, as real peaks should be much sharper than the ones drawn by lines.
  • Figures 12 and 13 have been reformulated with multiple frames, according to your suggestion.
  1. From the experimental point of view, did the authors measure optical transmission of samples under study? They all seem opaque from the picture. Is it not important for simple usage as protective layer?
  • The transmittance of the studied samples were measured and added to the experimental part and the results. Figure 4 has been added.
  1. Authors should better describe the preparation procedure for samples before measurements. Were they cut and polished? Though the optical-grade surface quality may not be required for gamma measurements, the accuracy of thickness determination depends on the surface type.
  • The preparation of samples for gamma measurements was explained in the section of gamma measurements. The samples were cut and polished before the measurement process, with thicknesses ranging from 4 to 5 mm.
  1. In section 2.2 the flux is stated to be 106, where it is 10^6 counts/s in the reference 17.
  • Done
  1. XRD results in Fig. 3 all show an amorphous halo, which is expected for glass samples. To me, the Fig. 3 may be omitted and replaced by a statement in the text.
  • XRD tests are necessary to show that the prepared glass is amorphous. This procedure requires showing the results of these tests through Figure 2.
  1. Values in Table 2 may be rounded to the third digit as the uncertainty is in the order of 0.001 - 0.003.
  • The values in Table 2 have been revised as your suggestion.
  1. I would add some extra references to the introduction, for instance P. Kaur et al. / Spectrochimica Acta Part A 206 (2019) 367–377 or . Limkitjaroenporn et al. / Journal of Physics and Chemistry of Solids 72 (2011) 245–25.
  • Your suggested reference has been added.

Thank you for reviewer.

Reviewer 2 Report

It seems a good piece of work

Author Response

Than you for reviewer

Reviewer 3 Report

My primary question on this manuscript is concerned with the proposed end use of the glasses. At various points the synthesised materials are compared with radiation tolerant optical glasses for shielding applications, such as the RS series from Schott or are compared to non-transparent materials such as barite concrete. However the reader is not presented with two important material characteristics for the glasses:

1) The optical transmission in the visible region of the spectrum

2) The maximum thickness that these glasses can be cast in.

Without this data the reader is left uncertain whether these glasses have a practical utility in the application are under discussion. From my visual interpretation of figure 1, these glasses would appear to have very limited value as an alternative to the Schott RS glasses. Without knowing the thickness in which they can be cast it is hard to judge whether they are a practical alternative to shielding concrete.

On page 2 it is stated that the addition of copper is beneficial in slowing down fast neutrons, but this is not justified; on what basis is this statement made?

There are a very large number of studies on bismuth-oxide based glasses, to a considerable extent motivated as the authors state by the need to find alternatives to high lead-content glasses. However it is not clear to me exactly what this study really adds to the extensive literature, perhaps if this was made more clear and my concerns which I state above were addressed then with these revisions this could be accepted for publication. Some revision to the English would also enhance the manuscript and I noted that the URL for reference [21] leads to a non-existent page.

Author Response

Comments and Suggestions for Authors

My primary question on this manuscript is concerned with the proposed end use of the glasses. At various points the synthesised materials are compared with radiation tolerant optical glasses for shielding applications, such as the RS series from Schott or are compared to non-transparent materials such as barite concrete. However the reader is not presented with two important material characteristics for the glasses:

1) The optical transmission in the visible region of the spectrum

2) The maximum thickness that these glasses can be cast in.

Without this data the reader is left uncertain whether these glasses have a practical utility in the application are under discussion. From my visual interpretation of figure 1, these glasses would appear to have very limited value as an alternative to the Schott RS glasses. Without knowing the thickness in which they can be cast it is hard to judge whether they are a practical alternative to shielding concrete.

  • The transmittance of the studied samples were measured and added to the experimental part and the results. Figure 4 has been added.
  • As for the thickness of the samples, they can be prepared in different thicknesses. The samples under study that were prepared had a thickness between 1.2 and 1 cm. It can also be prepared in different thicknesses as we want. It was cut, polished and made to thicknesses of 4 to 5 mm to Gamma measurements.

On page 2 it is stated that the addition of copper is beneficial in slowing down fast neutrons, but this is not justified; on what basis is this statement made?

The statement has been reviewed.

Neutron absorbers materials consists of different and many type of materials, including borated stainless steel, B4C/Al composite, amorphous alloys, and B/Al alloy. Neutron shielding materials contain polymer-based composites, high density concrete, heavy metals, paraffin, and other neutron shielding materials with additives, such as hexagonal boron nitride (h-BN), boron carbide (B4C), boric acid, colemanite, cadmium (Cd), silver, copper, gadolinium (Gd), and samarium oxide (Sm2O3) fillers. The shielding materials present the good neutron shielding performance efficiently, and the neutrons can be effectively shielded via elastic and inelastic scattering.

There are a very large number of studies on bismuth-oxide based glasses, to a considerable extent motivated as the authors state by the need to find alternatives to high lead-content glasses. However it is not clear to me exactly what this study really adds to the extensive literature, perhaps if this was made more clear and my concerns which I state above were addressed then with these revisions this could be accepted for publication. Some revision to the English would also enhance the manuscript and I noted that the URL for reference [21] leads to a non-existent page.

·         The prepared glass was compared with other materials used for gamma shielding, whether transparent materials such as glass or opaque materials such as barite concrete and others, and it gave very good results, better than the commercial materials used.·         It is possible to use the prepared glass under study for making gamma shielding to preserve radioactive materials in laboratories, nuclear medicine center, research centers and universities. Or in nuclear facilities and installations.·         The English language has been reviewed by native English speaker and we do our best linguistically.·         As for The URL for reference 23, you can copy and paste the URL of the reference in the Google browser.

Thank you for reviewer.

Comments and Suggestions for Authors

My primary question on this manuscript is concerned with the proposed end use of the glasses. At various points the synthesised materials are compared with radiation tolerant optical glasses for shielding applications, such as the RS series from Schott or are compared to non-transparent materials such as barite concrete. However the reader is not presented with two important material characteristics for the glasses:

1) The optical transmission in the visible region of the spectrum

2) The maximum thickness that these glasses can be cast in.

Without this data the reader is left uncertain whether these glasses have a practical utility in the application are under discussion. From my visual interpretation of figure 1, these glasses would appear to have very limited value as an alternative to the Schott RS glasses. Without knowing the thickness in which they can be cast it is hard to judge whether they are a practical alternative to shielding concrete.

  • The transmittance of the studied samples were measured and added to the experimental part and the results. Figure 4 has been added.
  • As for the thickness of the samples, they can be prepared in different thicknesses. The samples under study that were prepared had a thickness between 1.2 and 1 cm. It can also be prepared in different thicknesses as we want. It was cut, polished and made to thicknesses of 4 to 5 mm to Gamma measurements.

On page 2 it is stated that the addition of copper is beneficial in slowing down fast neutrons, but this is not justified; on what basis is this statement made?

The statement has been reviewed.

Neutron absorbers materials consists of different and many type of materials, including borated stainless steel, B4C/Al composite, amorphous alloys, and B/Al alloy. Neutron shielding materials contain polymer-based composites, high density concrete, heavy metals, paraffin, and other neutron shielding materials with additives, such as hexagonal boron nitride (h-BN), boron carbide (B4C), boric acid, colemanite, cadmium (Cd), silver, copper, gadolinium (Gd), and samarium oxide (Sm2O3) fillers. The shielding materials present the good neutron shielding performance efficiently, and the neutrons can be effectively shielded via elastic and inelastic scattering.

There are a very large number of studies on bismuth-oxide based glasses, to a considerable extent motivated as the authors state by the need to find alternatives to high lead-content glasses. However it is not clear to me exactly what this study really adds to the extensive literature, perhaps if this was made more clear and my concerns which I state above were addressed then with these revisions this could be accepted for publication. Some revision to the English would also enhance the manuscript and I noted that the URL for reference [21] leads to a non-existent page.

·         The prepared glass was compared with other materials used for gamma shielding, whether transparent materials such as glass or opaque materials such as barite concrete and others, and it gave very good results, better than the commercial materials used.·         It is possible to use the prepared glass under study for making gamma shielding to preserve radioactive materials in laboratories, nuclear medicine centers, research centers and universities. Or in nuclear facilities and installations.·         The English language has been reviewed by a native English speakers and we do our best linguistically.·         As for The URL for reference 23, you can copy and paste the URL of the reference in the Google browser.

Thank you for reviewer.

Round 2

Reviewer 1 Report

Authors responded to the reviewer comments. Manuscript was somewhat improved. It may be now published in the Inorganics journal.

Author Response

Thank you for reviewer

Reviewer 3 Report

Dear authors, thank you for your responses to my questions and for providing some additional data on the optical properties (transmittance) of the glasses in figure 4. However I do not understand how any passive (i.e. without a population inversion as in a laser) material can have a transmittance which exceeds 100%! Something has gone seriously wrong here with your data when plotted or perhaps even an instrumental error. I also find it very hard to believe that these glasses have any significant transmittance in the UV, let alone values approaching 40 to 50% as in your figure. A corrected version of this figure is essential, please also tell the reader whether the transmittance is "internal" or "external" (i.e. corrected for or not corrected for Fresnel reflectance at the surfaces) and what spectrophotometer was used to take the measurements.

Thank you for improving the quality of the English, that is appreciated.

Regarding thickness, my question was perhaps not clear enough, what I was asking, and you may not have an answer, is what is the maximum thickness that these glasses can have before crystalisation occurs. Knowing that they can be made up to about 1 cm thick is useful, do you know if that is the limit or just the maximum that was produced for this study?

Author Response

Dear authors, thank you for your responses to my questions and for providing some additional data on the optical properties (transmittance) of the glasses in figure 4. However I do not understand how any passive (i.e. without a population inversion as in a laser) material can have a transmittance which exceeds 100%! Something has gone seriously wrong here with your data when plotted or perhaps even an instrumental error. I also find it very hard to believe that these glasses have any significant transmittance in the UV, let alone values approaching 40 to 50% as in your figure.

The figure 4 has been reformulated.

A corrected version of this figure is essential, please also tell the reader whether the transmittance is "internal" or "external" (i.e. corrected for or not corrected for Fresnel reflectance at the surfaces) and what spectrophotometer was used to take the measurements.

. The recording of the transmittance values of a material can vary based on the application. Although the majority of industrial glasses give optical properties as external transmittance, values for filter glasses are usually given as internal transmittance. This is because anti-reflective coatings may be applied to filter glasses to avoid intensity losses from their surface. Therefore, the current glass deals with transmittance as external transmittance. 

The transmission spectra were recorded using a Shimadzu Ultraviolet Spectrometer 2700 in the 200-800 nm spectral range. page 3

Regarding thickness, my question was perhaps not clear enough, what I was asking, and you may not have an answer, is what is the maximum thickness that these glasses can have before crystalisation occurs. Knowing that they can be made up to about 1 cm thick is useful, do you know if that is the limit or just the maximum that was produced for this study?

Crystallization in glass does not depend on the thickness, but depends on the heat treatments that are done for the glass. And the process of annealing and casting and time is greatly influential in that. In general, crystallization phenomena in glass occur by different complex mechanisms which represent the way the structure is arranged from the amorphous to the crystalline state. These mechanisms are mainly governed by the heat gradients throughout the system during heat treatment. The starting glass composition has a direct influence on the viscosity of the system, which is a key parameter in crystal nucleation and growth.